# CRITICAL INITIALISATION IN CONTINUOUS APPROXIMATIONS OF BINARY NEURAL NETWORKS

**George Stamatescu, Ian Fuss and Langford B. White**
School of Electrical and Electronic Engineering
University of Adelaide
Adelaide, Australia
{george.stamatescu}@gmail.com
{lang.white,ian.fuss}@adelaide.edu.au

**Federica Gerace**
Institut de Physique Théorique
CNRS & CEA & Université Paris-Saclay
Saclay, France
federicagerace91@gmail.com

**Carlo Lucibello**
Bocconi Institute for DataScience and Analytics
Bocconi University
Milan, Italy
carlo.lucibello@unibocconi.it

## ABSTRACT

The training of stochastic neural network models with binary ($\pm 1$) weights and activations via continuous surrogate networks is investigated. We derive new surrogates using a novel derivation based on writing the stochastic neural network as a Markov chain. This derivation also encompasses existing variants of the surrogates presented in the literature. Following this, we theoretically study the surrogates at initialisation. We derive, using mean field theory, a set of scalar equations describing how input signals propagate through the randomly initialised networks. The equations reveal whether so-called critical initialisations exist for each surrogate network, where the network can be trained to arbitrary depth. Moreover, we predict theoretically and confirm numerically, that common weight initialisation schemes used in standard continuous networks, when applied to the mean values of the stochastic binary weights, yield poor training performance. This study shows that, contrary to common intuition, the means of the stochastic binary weights should be initialised close to $\pm 1$, for deeper networks to be trainable.

## 1  INTRODUCTION

The problem of learning with low-precision neural networks has seen renewed interest in recent years, in part due to the deployment of neural networks on low-power devices. Currently, deep neural networks are trained and deployed on GPUs, without the memory or power constraints of such devices. Binary neural networks are a promising solution to these problems. If one is interested in addressing memory usage, the precision of the weights of the network should be reduced, with the binary case being the most extreme. In order to address power consumption, networks with both binary weights and neurons can deliver significant gains in processing speed, even making it feasible to run the neural networks on CPUs Rastegari et al. (2016). Of course, introducing discrete variables creates challenges for optimisation, since the networks are not continuous and differentiable.

Recent work has opted to train binary neural networks directly via backpropagation on a differentiable surrogate network, thus leveraging automatic differentiation libraries and GPUs. A key to this approach is in defining an appropriate *differentiable* surrogate network as an approximation to the discrete model. A principled approach is to consider binary *stochastic* variables and use this stochasticity to "smooth out" the non-differentiable network. This includes the cases when (i) only weights, and (ii) both weights and neurons are stochastic and binary.

In this work we study two classes of surrogates, both of which make use of the Gaussian central limit theorem (CLT) at the receptive fields of each neuron. In either case, the surrogates are written

as differentiable functions of the continuous means of stochastic binary weights, but with more complicated expressions than for standard continuous networks.

One approximation, based on analytic integration, yields a class of deterministic surrogates Soudry et al. (2014). The other approximation is based on the local reparameterisation trick (LRT) Kingma & Welling (2013), which yields a class of stochastic surrogates Shayer et al. (2017). Previous works have relied on heuristics to deal with binary neurons Peters & Welling (2018), or not backpropagated gradients correctly. Moreover, none of these works considered the question of initialisation, potentially limiting performance.

The seminal papers of Saxe et al. (2013), Poole et al. (2016), Schoenholz et al. (2016) used a mean field formalism to explain the empirically well known impact of initialization on the dynamics of learning in standard networks. From one perspective the formalism studies how signals propagate forward and backward in wide, random neural networks, by measuring how the variance and correlation of input signals evolve from layer to layer, knowing the distributions of the weights and biases of the network. By studying these moments the authors in Schoenholz et al. (2016) were able to explain how heuristic initialization schemes avoid the "vanishing and exploding gradients problem" Glorot & Bengio (2010), establishing that for neural networks of arbirary depth to be trainable they must be initialised at "criticality", which corresponds to initial correlation being preserved to any depth.

The paper makes three contributions. The first contribution is the presentation of new algorithms, with a new derivation able to encompass both surrogates, and all choices of stochastic binary weights, or neurons. The derivation is based on representing the stochastic neural network as a Markov chain, a simplifying and useful development. As an example, using this representation we are easily able to extend the LRT to the case of stochastic binary neurons, which is new. This was not possible in Shayer et al. (2017), who only considered stochastic binary weights. As a second example, the deterministic surrogate of Soudry et al. (2014) is easily derived, without the need for Bayesian message passing arguments. Moreover, unlike Soudry et al. (2014) we correctly backpropagate through variance terms, as we discuss.

The second contribution is the theoretical analysis of both classes of surrogate at initialisation, through the prism of signal propagation theory Poole et al. (2016), Schoenholz et al. (2016). This analysis is achieved through novel derivations of the dynamic mean field equations, which hinges on the use of self-averaging arguments Mezard et al. (1987). The results of the theoretical study, which are supported by numerical simulations and experiment, establish that for a surrogate of arbitrary depth to be trainable, it must be randomly initialised at "criticality". In practical terms, criticality corresponds to using initialisations that avoid the "vanishing and exploding gradients problem" Glorot & Bengio (2010). We establish the following key results:

- For networks with stochastic binary weights and neurons, the deterministic surrogate can achieve criticality, while the LRT cannot.

- For networks with stochastic binary weights and continuous neurons, the LRT surrogate can achieve criticality (no deterministic surrogate exists for this case)

In both cases, the critical initialisation corresponds to randomly initialising the means of the binary weights close to $\pm 1$, a counter intuitive result.

A third contribution is the consideration of the signal propagation properties of random binary networks, in the context of training a differentiable surrogate network. We derive these results, which are partially known, and in order to inform our discussion of the experiments.

This paper provides insights into the dynamics and training of the class of binary neural network models. To date, the initialisation of any binary neural network algorithm has not been studied, although the effect of quantization levels has been explored through this perspective Blumenfeld et al. (2019). Currently, the most popular surrogates are based on the so-called "Straight-Through" estimator Bengio et al. (2013), which relies on heuristic definitions of derivatives in order to define a gradient. However, this surrogate typically requires the use of batch normalization, and other heuristics. The contributions in this paper may help shed light on what is holding back the more principled algorithms, by suggesting practical advice on how to initialise, and what to expect during training.

**Paper outline:** In section 2 we present the binary neural network algorithms considered. In subsection 2.1 we define binary neural networks and subsection 2.2 their stochastic counterparts. In subsection 2.3 we use these definitions to present new and existing surrogates in a coherent framework, using the Markov chain representation of a neural network to derive variants of both the deterministic surrogate, and the LRT-based surrogates. We derive the LRT for the case of stochastic binary weights, and both LRT and deterministic surrogates for the case of stochastic binary weights and neurons. In section 3 we derive the signal propagation equations for both the deterministic and stochastic LRT surrogates. This includes deriving the explicit depth scales for trainability, and solving the equations to find the critical initialisations for each surrogate, if they exist. In section 4 we present the numerical simulations of wide random networks, to validate the mean field description, and experimental results to test the trainability claims. In section 5 we summarize the key results, and provide a discussion of the insights they provide.

## 2 BINARY NEURAL NETWORK ALGORITHMS

### 2.1 CONTINUOUS NEURAL NETWORKS AND BINARY NEURAL NETWORKS

A neural network model is typically defined as a deterministic non-linear function. We consider a fully connected feedforward model, which is composed of $N^\ell \times N^{\ell-1}$ weight matrices $W^\ell$ and bias vectors $b^\ell$ in each layer $\ell \in \{1, \ldots, L\}$, with elements $W_{ij}^\ell \in \mathbb{R}$ and $b_i^\ell \in \mathbb{R}$. Given an input vector $x^0 \in \mathbb{R}^{N_0}$, the network is defined in terms of the following recursion,

$$x^\ell = \phi^\ell(h^\ell), \qquad h^\ell = \frac{1}{\sqrt{N^{\ell-1}}} W^\ell x^{\ell-1} + b^\ell \tag{1}$$

where the pointwise non-linearity is, for example, $\phi^\ell(\cdot) = \max(0, \cdot)$. We refer to the input to a neuron, such as $h^\ell$, as the pre-activation field.

A deterministic binary neural network simply has weights $W_{ij}^\ell \in \{\pm 1\}$ and $\phi^\ell(\cdot) = \text{sign}(\cdot)$, and otherwise the same propagation equations. Of course, this is not differentiable, thus we instead consider stochastic binary variables in order to smooth out the non-differentiable network. Ideally, the product of training a surrogate of a stochastic binary network is a deterministic (or stochastic) binary network that is able to generalise from its training set.

### 2.2 STOCHASTIC BINARY NEURAL NETWORKS

In stochastic binary neural networks we denote the matrices as $\mathbf{S}^\ell$ with all weights[1] $\mathbf{S}_{ij}^\ell \in \{\pm 1\}$ being independently sampled binary variables with probability is controlled by the mean $M_{ij}^\ell = \mathbb{E}\mathbf{S}_{ij}^\ell$. Neuron activation in this model are also binary random variables, due to pre-activation stochasticity and to inherent noise. We consider parameterised neurons such that the mean activation conditioned on the pre-activation is given by some function taking values in $[-1, 1]$, i.e. $\mathbb{E}[\mathbf{x}_i^\ell \,|\, \mathbf{h}_i^\ell] = \phi(\mathbf{h}_i^\ell)$, for example $\phi(\cdot) = \tanh(\cdot)$. We write the propagation rules for the stochastic network as follows:

$$\mathbf{S}^\ell \sim p(\bullet; M^\ell); \qquad \mathbf{h}^\ell = \frac{1}{\sqrt{N^{\ell-1}}} \mathbf{S}^\ell \mathbf{x}^{\ell-1} + b^\ell; \qquad \mathbf{x}^\ell \sim p(\bullet; \phi(\mathbf{h}^\ell)) \tag{2}$$

Notice that the distribution of $\mathbf{x}^\ell$ factorizes when conditioning on $\mathbf{x}^{\ell-1}$. The form of the neuron's mean function $\phi(\cdot)$ depends on the underlying noise model. We can express a binary random variable $\mathbf{x} \in \{\pm 1\}$ with $\mathbf{x} \sim p(\mathbf{x}; \theta)$ via its latent variable formulation $\mathbf{x} = \text{sign}(\theta + \alpha \mathbf{L})$. In this form $\theta$ is referred to as a "natural" parameter, and the term $\mathbf{L}$ is a latent random noise, whose cumulative distribution function $\sigma(\cdot)$ determines the form of the non-linearity since $\phi(\cdot) = 2\sigma(\cdot) - 1$. In general the form of $\phi(\cdot)$ will impact on the surrogates' performance, including within and beyond the mean field description presented here. However, a result from the analysis in Section 3 is that choosing a deterministic binary neuron, ie. the $\text{sign}(\cdot)$ function, or a stochastic binary neuron, produces the same signal propagation equations, up to a scaling constant.

---

[1]We denote random variables with bold font. Also, following physics' jargon, we refer to binary $\pm 1$ variables as Ising spins or just spins.

## 2.3 DERIVATIONS OF NEW AND EXISTING SURROGATE NETWORKS

The idea behind several recent papers Soudry et al. (2014), Baldassi et al. (2018), Shayer et al. (2017), Peters & Welling (2018) is to adapt the mean of the binary stochastic weights, with the stochastic model essentially used to "smooth out" the discrete variables and arrive at a differentiable function, open to the application of continuous optimisation techniques. We now derive both the deterministic surrogate and LRT-based surrogates, in a common framework. We consider a supervised classification task, with training set $\mathcal{D} = \{x_\mu, y_\mu\}_{\mu=1}^P$, with $y_\mu$ the label. we define a loss function for our surrogate model via

$$\mathcal{L}(M, b) = -\frac{1}{P} \sum_{\mu=1}^P \log \mathbb{E}_{\mathbf{S}, \mathbf{x}} \, p(y_\mu \,|\, x_\mu, \mathbf{S}, \mathbf{x}, b), \tag{3}$$

For a given input $x_\mu$ and a realization of weights, neuron activations and biases in all layers, denoted by $(\mathbf{S}, \mathbf{x}, b)$, the stochastic neural network produces a probability distribution over the classes. Expectations over weights and activations are given by the mean values, $\mathbb{E}\mathbf{S}^\ell = M^\ell$ and $\mathbb{E}[\mathbf{x}^\ell | \mathbf{h}^\ell] = \phi(\mathbf{h}^\ell)$. This objective can be recognised as a (minus) marginal likelihood, thus this method could be described as Type II maximum likelihood, or empirical Bayes.

The starting point for our derivations comes from rewriting the expectation equation 3 as the marginalization of a Markov chain, with layers $\ell$ indexes corresponding to time indices.

**Markov chain representation of stochastic neural network:**

$$\mathbb{E}_{\mathbf{S}, \mathbf{x}} \, p(y_\mu \,|\, x_\mu, \mathbf{S}, b, \mathbf{x}) = \sum_{\mathbf{S}, \mathbf{x} \,:\, \mathbf{x}^0 = x_\mu} p(y_\mu \,|\, \mathbf{x}^L) \prod_{\ell=1}^L p(\mathbf{x}^\ell \,|\, \mathbf{x}^{\ell-1}, \mathbf{S}^\ell) \, p(\mathbf{S}^\ell; M^\ell)$$

$$= \sum_{\mathbf{S}^L, \mathbf{x}^{L-1}} p(y_\mu | \mathbf{S}^L, \mathbf{x}^{L-1}) p(\mathbf{S}^L) \sum_{\mathbf{S}^{L-1}, \mathbf{x}^{L-2}} p(\mathbf{x}^{L-1} | \mathbf{x}^{L-2}, \mathbf{S}^{L-1}) p(\mathbf{S}^{L-1}) \cdots \sum_{\mathbf{S}^1} p(\mathbf{x}^1 | x_\mu, \mathbf{S}^1) p(\mathbf{S}^1) \tag{4}$$

where in the second line we dropped from the notation $p(\mathbf{S}^\ell; M^\ell)$ the dependence on $M^\ell$ for brevity. Therefore, for a stochastic network the forward pass consists in the propagation of the joint distribution of layer activations, $p(\mathbf{x}^\ell | x_\mu)$, according to the Markov chain. We drop the explicit dependence on the initial input $x_\mu$ from now on.

In what follows we will denote with $\phi(\mathbf{h}^\ell)$ the average value of $\mathbf{x}^\ell$ according to $p(\mathbf{x}^\ell)$. The first step to obtaining a differentiable surrogate is to introduce continuous random variables. We take the limit of large layer width and appeal to the central limit theorem to model the field $\mathbf{h}^\ell$ as Gaussian, with mean $\bar{h}^\ell$ and covariance matrix $\Sigma^\ell$.

**Assumption 1: (CLT for stochastic binary networks)** *In the large $N$ limit, under the Lyapunov central limit theorem, the field $\mathbf{h}^\ell = \frac{1}{\sqrt{N^{\ell-1}}} \mathbf{S}^\ell \mathbf{x}^{\ell-1} + b^\ell$ converges to a Gaussian random variable with mean $\bar{h}_i^\ell = \frac{1}{\sqrt{N^{\ell-1}}} \sum_j M_{ij}^\ell \phi(\mathbf{h}_j^{\ell-1}) + b_i^\ell$ and covariance matrix $\Sigma^\ell$ with diagonal $\Sigma_{ii}^\ell = \frac{1}{N^{\ell-1}} \sum_j 1 - (M_{ij}^\ell \phi(\mathbf{h}_j^{\ell-1}))^2$.*

While this assumption holds true for large enough networks, due to $\mathbf{S}^\ell$ and $\mathbf{x}^{\ell-1}$ independency, the Assumption 2 below, is stronger and tipically holds only at initialization.

**Assumption 2: (correlations are zero)** *We assume the independence of the pre-activation field $\mathbf{h}^\ell$ between any two dimensions. Specifically, we assume the covariance $\Sigma = Cov(\mathbf{h}^\ell, \mathbf{h}^\ell)$ to be well approximated by $\Sigma_{MF}^\ell(\phi(\mathbf{h}^{\ell-1}))$, with MF denoting the mean field (factorized) assumption, where*

$$\left( \Sigma_{MF}^\ell(x) \right)_{ii'} = \delta_{ii'} \frac{1}{N^{\ell-1}} \sum_j 1 - (M_{ij}^\ell \phi(\mathbf{h}_j^{\ell-1}))^2 \tag{5}$$

This assumption approximately holds assuming the neurons in each layer are not strongly correlated. In the first layer this is certainly true, since the input neurons are not random variables[2]. In subsequent layers, since the fields $\mathbf{h}_i^\ell$ and $\mathbf{h}_j^\ell$ share stochastic neurons from the previous layer, this cannot be assumed to be true. We expect this correlation to not play a significant role, since

---

[2]In this case the variance is actually $\frac{1}{N^{\ell-1}} \sum_j \left( 1 - (M_{ij}^1)^2 \right) (x_{\mu,j})^2$.

the weights act to decorrelate the fields, and the neurons are independently sampled. However, the choice of surrogate influences the level of dependence. The sampling procedure used within the local reparametrization trick reduces correlations since variables are sampled, while the deterministic surrogate entirely discards them.

We obtain either surrogate model by successively approximating the marginal distributions, $p(\mathbf{x}^\ell) = \int d\mathbf{h}^\ell \, p(\mathbf{x}^\ell|\mathbf{h}^\ell) \approx \hat{p}(\mathbf{x}^\ell)$, starting from the first layer. We can do this by either (i) marginalising over the Gaussian field using analytic integration, or (ii) sampling from the Gaussian. After this, we use the approximation $\hat{p}(\mathbf{x}_i^\ell)$ to form the Gaussian approximation for the next layer, and so on.

**Deterministic surrogate:** We perform the analytic integration based on the analytic form of $p(\mathbf{x}_i^{\ell+1}|\mathbf{h}^\ell) = \sigma(\mathbf{x}_i^\ell \mathbf{h}_i^\ell)$, with $\sigma(\cdot)$ a sigmoidal function. In the case that $\sigma(\cdot)$ is the Gaussian CDF, we obtain $\hat{p}(\mathbf{x}_i^\ell)$ exactly[3] by the Gaussian integral of the Gaussian cumulative distribution function,

$$\hat{p}(\mathbf{x}_i^\ell) = \int dh \, \sigma(\mathbf{x}_i^\ell h) \, \mathcal{N}(h \, ; \bar{h}_i^\ell, \Sigma_{MF,ii}^\ell) = \Phi\left(\frac{\bar{h}_i^\ell}{(1 + \Sigma_{MF}^\ell)_{ii}^{1/2}} \mathbf{x}_i^\ell\right) \tag{6}$$

Since we start from the first layer, all random variables are marginalised out, and thus $\bar{h}_i^\ell$ has no dependence on random $\mathbf{h}_j^{\ell-1}$ via the neuron means $\phi(\mathbf{h}^\ell)$ as in Assumption 1. Instead, we have dependence on means $\bar{x}^\ell = \mathbb{E}_{\mathbf{h}^\ell} \mathbb{E}\left[\mathbf{x}^\ell \,|\, \mathbf{h}^\ell\right] = \mathbb{E}_{\mathbf{h}^\ell} \phi(\mathbf{h}^\ell)$. Thus it is convenient to define the mean under $\hat{p}(\mathbf{x}_i^\ell)$ as $\varphi^\ell(\bar{h}, \sigma^2) = \int dh \, \phi^\ell(h) \, \mathcal{N}(h \, ; \bar{h}, \sigma^2)$. In the case that $\sigma(\cdot)$ is the Gaussian CDF, then $\varphi^\ell(\cdot)$ is the error function. Finally, the forward pass can be expressed as

$$\bar{x}^\ell = \varphi^\ell(h^\ell) \qquad h^\ell = (1 + \Sigma_{MF}^\ell)^{-\frac{1}{2}} \bar{h}^\ell \qquad \bar{h}^\ell = \frac{1}{\sqrt{N^{\ell-1}}} M^\ell \bar{x}^{\ell-1} + b^\ell, \tag{7}$$

This is a more general formulation than that in Soudry et al. (2014), which considered sign activations, which we obtain in the appendices as a special case. Furthermore, in all implementations we backpropagate through the variance terms $\Sigma_{MF}^{-\frac{1}{2}}$, which were ignored in the previous work of Soudry et al. (2014). Note that the derivation here is simpler as well, not requiring complicated Bayesian message passing arguments, and approximations therein.

**LRT surrogate:** The basic idea here is to rewrite the incoming Gaussian field $\mathbf{h} \sim \mathcal{N}(\mu, \Sigma)$ as $\mathbf{h} = \mu + \sqrt{\Sigma}\,\boldsymbol{\epsilon}$ where $\boldsymbol{\epsilon} \sim \mathcal{N}(0, I)$. Thus expectations over $\mathbf{h}$ can be written as expectations over $\boldsymbol{\epsilon}$ and approximated by sampling. The resulting network is thus differentiable, albeit not deterministic. The forward propagation equations for this surrogate are

$$\mathbf{h}^\ell = \frac{1}{\sqrt{N^{\ell-1}}} M^\ell \bar{\mathbf{x}}^{\ell-1} + b^\ell + \sqrt{\Sigma_{MF}^\ell(\bar{\mathbf{x}}^{\ell-1})}\,\boldsymbol{\epsilon}^\ell, \qquad \bar{\mathbf{x}}^\ell = \phi^\ell(\mathbf{h}^\ell). \tag{8}$$

The local reparameterisation trick (LRT) Kingma & Welling (2013) has been previously used to obtain differentiable surrogates for binary networks. The authors of Shayer et al. (2017) considered only the case of stochastic binary weights, since they did not write the network as a Markov chain. Peters & Welling (2018) considered stochastic binary weights and neurons, but relied on other approximations to deal with the neurons, having not used the Markov chain representation.

The result of each approximation, applied successively from layer to layer by either propagating means and variances or by, produces a differentiable function of the parameters $M_{ij}^\ell$. It is then possible to perform gradient descent with respect to the $M$ and $b$. Ideally, at the end of training we obtain a binary network that attains good performance. This network could be a stochastic network, where we sample all weights and neurons, or a deterministic binary network. A deterministic network might be chosen taking the most likely weights, therefore setting $W_{ij}^\ell = \mathrm{sign}(M_{ij}^\ell)$, and replacing the stochastic neurons with $\mathrm{sign}(\cdot)$ activations.

## 3 Signal propagation theory for continuous surrogates

Since all the surrogates still retain the basic neural network structure of layerwise processing, crucially applying backpropagation for optimisation, it is reasonable to expect that surrogates are likely

---

[3]In the Appendices we show that other sigmoidal $\sigma(\cdot)$ can be approximated by a Gaussian CDF.

to inherit similar "training problems" as standard neural networks. In this section we apply this formalism to the surrogates considered, given *random* initialisation of the means $M_{ij}^\ell$ and biases $b_i^\ell$. We are able to solve for the conditions of critical initialisation for each surrogate, which essentially allow signal to propagate forwards, and gradients to propagate backwards, without the effects such as neuron saturation. The critical initialisation for the surrogates, the key results of the paper, are provided in Claims 1 and 3.

## 3.1 FORWARD SIGNAL PROPAGATION FOR STANDARD CONTINUOUS NETWORKS

We first recount the formalism developed in Poole et al. (2016). Assume the weights of a standard continuous network are initialised with $W_{ij}^\ell \sim \mathcal{N}(0, \sigma_w^2)$, biases $b^\ell \sim \mathcal{N}(0, \sigma_b^2)$, and input signal $x_a^0$ has zero mean $\mathbb{E} x^0 = 0$ and variance $\mathbb{E}[x_a^0 \cdot x_a^0] = q_{aa}^0$, and with $a$ denoting a particular input pattern. As before, the signal propagates via Equation 1 from layer to layer.

We are interested in computing, from layer to layer, the variance $q_{aa}^\ell = \frac{1}{N_\ell} \sum_i (h_{i;a}^\ell)^2$ from a particular input $x_a^0$, and also the covariance between the pre-activations $q_{ab}^\ell = \frac{1}{N_\ell} \sum_i h_{i;a}^\ell h_{i;b}^\ell$, arising from two different inputs $x_a^0$ and $x_b^0$ with given covariance $q_{ab}^0$. The mean field approximation used here replaces each element in the pre-activation field $h_i^\ell$ by a Gaussian random variable whose moments are matched. Assuming also independence within a layer; $\mathbb{E} h_{i;a}^\ell h_{j;a}^\ell = q_{aa}^\ell \delta_{ij}$ and $\mathbb{E} h_{i;a}^\ell h_{j;b}^\ell = q_{ab}^\ell \delta_{ij}$, one can derive recurrence relations from layer to layer,

$$q_{aa}^\ell = \sigma_w^2 \int Dz \phi^2(\sqrt{q_{aa}^{\ell-1}} z) + \sigma_b^2 = \sigma_w^2 \mathbb{E} \phi^2(h_{j,a}^{\ell-1}) + \sigma_b^2 \tag{9}$$

with $Dz = \frac{dz}{\sqrt{2\pi}} e^{-\frac{z^2}{2}}$ the standard Gaussian measure. The recursion for the covariance is given by

$$q_{ab}^\ell = \sigma_w^2 \int Dz_1 Dz_2 \phi(u_a)\phi(u_b) + \sigma_b^2 = \sigma_w^2 \mathbb{E}\big[\phi(h_{j,a}^{\ell-1})\phi(h_{j,b}^{\ell-1})\big] + \sigma_b^2 \tag{10}$$

where $u_a = \sqrt{q_{aa}^{\ell-1}} z_1$, $u_b = \sqrt{q_{bb}^{\ell-1}}\big(c_{ab}^{\ell-1} z_1 + \sqrt{1 - (c_{ab}^{\ell-1})^2} z_2\big)$, and we identify $c_{ab}^\ell$ as the correlation in layer $\ell$. The other important quantity is the slope of the correlation recursion equation or mapping from layer to layer, denoted as $\chi$, which is given by:

$$\chi = \frac{\partial c_{ab}^\ell}{\partial c_{ab}^{\ell-1}} = \sigma_w^2 \int Dz_1 Dz_2 \, \phi'(u_a)\phi'(u_b) \tag{11}$$

We denote $\chi$ at the fixed point $c^* = 1$ as $\chi_1$. As discussed Poole et al. (2016), when $\chi_1 = 1$, correlations can propagate to arbitrary depth.

**Definition 1:** *Critical initialisations are the points $(\sigma_b^2, \sigma_w^2)$ corresponding to $\chi_1 = 1$.*

Furthermore, $\chi_1$ is equivalent to the mean square singular value of the Jacobian matrix for a single layer $J_{ij} = \frac{\partial h_i^\ell}{\partial h_j^{\ell-1}}$, as explained in Poole et al. (2016). Therefore controlling $\chi_1$ will prevent the gradients from either vanishing or growing exponentially with depth. We thus define critical initialisations as follows. This definition also holds for the surrogates which we now study.

## 3.2 SIGNAL PROPAGATION THEORY FOR DETERMINISTIC SURROGATES

For the deterministic surrogate model we assume at initialization that the binary weight means $M_{ij}^\ell$ are drawn independently and identically from a distribution $P(M)$, with mean zero and variance of the means given by $\sigma_m^2$. For instance, a valid distribution could be a clipped Gaussian[4], or another stochastic binary variable, for example $P(M) = \frac{1}{2}\delta(M + \sigma_m) + \frac{1}{2}\delta(M - \sigma_m)$, whose variance is $\sigma_m^2$. The biases at initialization are distributed as $b^\ell \sim \mathcal{N}(0, \sigma_b^2)$.

We show in Appendix B that the stochastic and deterministic binary neuron cases reduce to the same signal propagation equations, up to scaling constants. In light of this, we consider the deterministic

---

[4]That is, sample from a Gaussian then pass the sample through a function bounded on the interval $[-1, 1]$.

sign($\cdot$) neuron case, since equation for the field is slightly simpler:

$$h_i^\ell = \frac{\sum_j M_{ij}^\ell \varphi(h_j^{\ell-1}) + \sqrt{N^{\ell-1}} b_i^\ell}{\sqrt{\sum_j [1 - (M_{ij}^\ell)^2 \varphi^2(h_j^{\ell-1})]}} \tag{12}$$

which we can be read from the Eq. 7. As in the continuous case we are interested in computing the variance $q_{aa}^\ell = \frac{1}{N_\ell} \sum_i (h_{i;a}^\ell)^2$ and covariance $\mathbb{E} h_{i;a}^\ell h_{j;b}^\ell = q_{ab}^\ell \delta_{ij}$, via recursive formulae. The key to the derivation is recognising that the denominator $\sqrt{\Sigma_{MF,ii}^\ell}$ is a self-averaging quantity Mezard et al. (1987). This means it concentrates in probability to its expected value for large $N$. Therefore we can safely replace it with its expectation. Following this self-averaging argument, we can take expectations more readily as shown in the appendices. We find the variance recursion to be

$$q_{aa}^\ell = \frac{\sigma_m^2 \mathbb{E}\varphi^2(h_{j,a}^{l-1}) + \sigma_b^2}{1 - \sigma_m^2 \mathbb{E}\varphi^2(h_{j,a}^{l-1})} \tag{13}$$

Based on this expression, and assuming $q_{aa} = q_{bb}$, the correlation recursion can be written as

$$c_{ab}^\ell = \frac{1 + q_{aa}^\ell}{q_{aa}^\ell} \frac{\sigma_m^2 \mathbb{E}\varphi(h_{j,a}^{l-1})\varphi(h_{j,b}^{l-1}) + \sigma_b^2}{1 + \sigma_b^2} \tag{14}$$

The slope of the correlation mapping from layer to layer, when the normalized length of each input is at its fixed point $q_{aa}^\ell = q_{bb}^\ell = q^*(\sigma_m, \sigma_b)$, denoted as $\chi$, is given by:

$$\chi = \frac{\partial c_{ab}^\ell}{\partial c_{ab}^{\ell-1}} = \frac{1 + q^*}{1 + \sigma_b^2} \sigma_m^2 \int Dz_1 Dz_2 \varphi'(u_a)\varphi'(u_b) \tag{15}$$

where $u_a$ and $u_b$ are defined exactly as in the continuous case. Refer to the appendices for full details of the derivation.

### 3.2.1 CRITICAL INITIALISATION: DETERMINISTIC SURROGATE

The condition for critical initialisation is $\chi_1 = 1$, since this determines the stability of the correlation map fixed point $c^* = 1$. Note that for the deterministic surrogate this is always a fixed point. We can solve for the hyper-parameters $(\sigma_b^2, \sigma_m^2)$ that satisfy this condition, using the dynamical equations of the network.

**Claim 1:** *The points $(\sigma_b^2, \sigma_m^2)$ corresponding to critical initialisation are given by $\sigma_m^2 = 1/\mathbb{E}[(\varphi'(\sqrt{q^*}z))^2] + \mathbb{E}[\varphi^2(\sqrt{q^*}z)]$ and finding $\sigma_b^2$ that satisfies*

$$q_{aa}^\ell = \sigma_b^2 + (\sigma_b^2 + 1)\frac{\mathbb{E}\varphi^2(h_{j,a}^{l-1})}{\mathbb{E}[(\varphi'(\sqrt{q^*}z))^2]}$$

This can be established by rearranging Equations 13 and 15. We solve for $\sigma_b^2$ numerically, as shown in Figure 3, for different neuron noise models and hence non-linearities $\varphi(\cdot)$. We find that the critical initialisation for any of these design choices is close to the point $(\sigma_m^2, \sigma_b^2) = (1, 0)$. However, it is not just the singleton point, as for example in Hayou et al. (2019) for the ReLu case for standard networks. We plot the solutions in the Appendix.

### 3.2.2 ASYMPTOTIC EXPANSIONS AND DEPTH SCALES

The depth scales, as derived in Schoenholz et al. (2016) provide a quantitative indicator to the number of layers correlations will survive for, and thus how trainable a network is. Similar depth scales can be derived for these deterministic surrogates. Asymptotically in network depth $\ell$, we expect that $|q_{aa}^\ell - q^*| \sim \exp(-\frac{\ell}{\xi_q})$ and $|c_{ab}^\ell - c^*| \sim \exp(-\frac{\ell}{\xi_c})$, where the terms $\xi_q$ and $\xi_c$ define the depth scales over which the variance and correlations of signals may propagate. We are most interested in the correlation depth scale, since it relates to $\chi$. The derivation is identical to that of Schoenholz et al. (2016). One can expand the correlation $c_{ab}^\ell = c^* + \epsilon^\ell$, and assuming $q_{aa}^\ell = q^*$, it is possible to write

$$\epsilon^{\ell+1} = \epsilon^\ell \Big[ \frac{1 + q^*}{1 + \sigma_b^2} \sigma_m^2 \int Dz \varphi'(u_1)\varphi'(u_2) + \mathcal{O}((\epsilon^\ell)^2) \tag{16}$$

The depth scale $\xi_c^{-1}$ are given by the log ratio $\log \frac{\epsilon^{\ell+1}}{\epsilon^\ell}$.

$$\xi_c^{-1} = -\log \left[ \frac{1+q^*}{1+\sigma_b^2} \sigma_m^2 \int Dz \varphi'(u_1)\varphi'(u_2) \right] = -\log \chi \tag{17}$$

We plot this depth scale in Figure 2. We derive the variance depth scale in the appendices, since it is different to the standard continuous case, but not of prime practical importance.

## 3.3 SIGNAL PROPAGATION THEORY FOR LOCAL REPARAMETERIZATION TRICK SURROGATES

From Equation 8, the pre-activation field for the perturbed surrogate with both stochastic binary weights and neurons is given by,

$$h_{i,a}^l = \frac{1}{\sqrt{N}} \sum_j M_{ij}^l \phi(h_{j,a}^{l-1}) + b_i^l + \epsilon_{i,a}^\ell \frac{1}{\sqrt{N}} \sqrt{\sum_j 1 - (M_{ij}^l)^2 \phi^2(h_{j,a}^{l-1})} \tag{18}$$

where we recall that $\epsilon \sim \mathcal{N}(0,1)$. The non-linearity $\phi(\cdot)$ can of course be derived from any valid binary stochastic neuron model. Appealing to the same self-averaging arguments used in the previous section, we find the variance map to be

$$q_{aa}^\ell = \mathbb{E}\left[(h_{i,a}^l)^2\right] = \sigma_m^2 \mathbb{E}\phi^2(h_{j,a}^{l-1}) + \sigma_b^2 + (1 - \sigma_m^2 \mathbb{E}\phi^2(h_{j,a}^{l-1})) = 1 + \sigma_b^2 \tag{19}$$

Interestingly, we see that the variance map does not depend on the variance of the means of the binary weights. This is not immediately obvious from the pre-activation field definition. In the covariance map we do not have such a simplification since the perturbation $\epsilon_{i,a}$ is uncorrelated between inputs $a$ and $b$. Thus the correlation map is given by

$$c_{ab}^l = \frac{\sigma_m^2 \mathbb{E}\phi(h_{j,a}^{l-1})\phi(h_{j,a}^{l-1}) + \sigma_b^2}{1 + \sigma_b^2} \tag{20}$$

## 3.4 CRITICAL INITIALISATION: LRT SURROGATES

**Claim 2:** *There is no critical initialisation for the local reparameterisation trick based surrogate, for a network with binary weights and neurons.*

*Proof:* The conditions for a critical initialisation are that $c^* = 1$ to be a fixed point and $\chi_1 = 1$. No such fixed point exists. We have a fixed point $c^* = 1$ if and only if $\sigma_m^2 = 1/\mathbb{E}[\phi^2(h_{j,a}^{l-1})]$. Note that $\sigma_m^2 \leq 1$. For any $\phi(z)$ which is the mean of the stochastic binary neuron, the expectation $\mathbb{E}[\phi^2(z)] \leq 1$. For example, consider $\phi(z) = \tanh(\kappa z)$ for any finite kappa.

We also considered the LRT surrogate with continuous $(\tanh(\cdot))$ neurons and stochastic binary weights. The derivations are very similar to the previous case, as we show in the appendix. The variance and correlation maps are given by

$$q_{aa}^\ell = \mathbb{E}\phi^2(h_{j,a}^{l-1})) + \sigma_b^2 \qquad c_{ab}^l = \frac{\sigma_m^2 \mathbb{E}\phi(h_{j,a}^{l-1})\phi(h_{j,a}^{l-1}) + \sigma_b^2}{\mathbb{E}\phi^2(h_{j,a}^{l-1}) + \sigma_b^2} \tag{21}$$

This leads to the following result,

**Claim 3:** *The critical initialisation for the LRT surrogate, for the case of continuous $\tanh(\cdot)$ neurons and stochastic binary weights is the singleton $(\sigma_b^2, \sigma_m^2) = (0, 1)$.*

*Proof:* From the correlation map we have a fixed point $c^* = 1$ if and only if $\sigma_m^2 = 1$, by inspection. In turn, the critical initialisation condition $\chi_1 = 1$ holds if $\mathbb{E}[(\phi'(h_{j,a}^{l-1}))^2] = \frac{1}{\sigma_m^2} = 1$. Thus, to find the critical initialisation, we need to find a value of $q_{aa} = \mathbb{E}\phi^2(h_{j,a}^{l-1}) + \sigma_b^2$ that satisfies this final condition. In the case that $\phi(\cdot) = \tanh(\cdot)$, then the function $(\phi'(h_{j,a}^{l-1}))^2 \leq 1$, taking the value 1 at the origin only, this requires $q_{aa} \to 0$. Thus we have the singleton $(\sigma_b^2, \sigma_m^2) = (0, 1)$ as the solution.

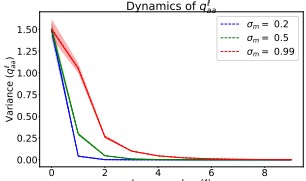 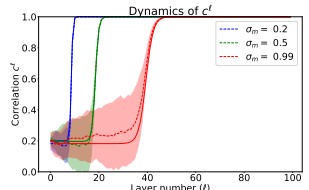 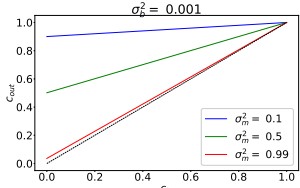

Figure 1: Dynamics of the variance and correlation maps, with simulations of a network of width $N = 1000$, 50 realisations, for various hyperparameter settings: $\sigma_m^2 \in \{0.2, 0.5, 0.99\}$ (blue, green and red respectively). (a) variance evolution, (b) correlation evolution. (c) correlation mapping ($c_{in}$ to $c_{out}$), with $\sigma_b^2 = 0.001$

## 4 NUMERICAL AND EXPERIMENTAL RESULTS

### 4.1 SIMULATIONS

We first verify that the theory accurately predicts the average behaviour of randomly initialised networks. We present simulations for the deterministic surrogate in Figure 1. We see that the average behaviour of random networks are well predicted by the mean field theory. Estimates of the variance and correlation are plotted, with dotted lines corresponding to empirical means and the shaded area corresponding to one standard deviation. Theoretical predictions are given by solid lines, with strong agreement for even finite networks. Similar plots can be produced for the LRT surrogate. In Appendix D we plot the depth scales as functions of $\sigma_m$ and $\sigma_b$.

### 4.2 TRAINING PERFORMANCE FOR DIFFERENT MEAN INITIALISATION $\sigma_m^2$

Here we experimentally test the predictions of the mean field theory by training networks to overfit a dataset in the supervised learning setting, having arbitrary depth and different initialisations. We consider first the performance of the deterministic and LRT surrogates, not their corresponding binary networks.

We use the MNIST dataset with reduced training set size (50%) and record the training performance (percentage of the training set correctly labeled) after 10 epochs of gradient descent over the training set, for various network depths $L < 70$ and different mean variances $\sigma_m^2 \in [0, 1)$. The optimizer used was SGD with Adam Kingma & Ba (2014) with a learning rate of $2 \times 10^{-4}$ chosen after simple grid search, and a batch size of 64. We see that the experimental results match the correlation depth scale derived, which are overlaid as dotted curves. A proportion of $3\xi_c$ was found to indicate the maximum attenuation in signal strength before trainability becomes difficult, similarly to previous works Schoenholz et al. (2016).

A reason we see the trainability not diverging in Figure 2 is that training time increases with depth, on top of requiring smaller learning rates for deeper networks, as described in Saxe et al. (2013). The experiment here used the same number of epochs regardless of depth, meaning shallower networks actually had an advantage over deeper networks. Note that the theory does not specify for how many steps of training the effects of critical initialisation will persist. Therefore, the number of steps we trained the network for is an arbitrary choice, and thus the experiments validate the theory in a more qualitative way. Results were similar for other optimizers, including SGD, SGD with momentum, and RMSprop. Note that these networks were trained without dropout, batchnorm or any other heuristics.

In Figure 2 we present the training performance for the deterministic surrogate and its stochastic binary counterpart. The results for a deterministic binary network were similar to a single Monte Carlo sample. Once again, we test our algorithms on the MNIST dataset and plot results after 5 epochs. We see that the performance of the stochastic network matches more closely the performance of the continuous surrogate as the number of samples increases, from $N = 5$ to $N = 100$ samples. We can report that the number of samples necessary to achieve better classification, at least for more shallow networks, appears to depends on the number of training epochs. This is a sensible relationship, since during the course of training we expect the means of the weights to polarise, moving closer to

the bounds $\pm 1$. Likewise, we expect that neurons, which initially have zero mean pre-activations, will also "saturate" during training, becoming either always "on" $(+1)$ or "off" $(-1)$. A stochastic network being "closer" to deterministic would require fewer samples overall.

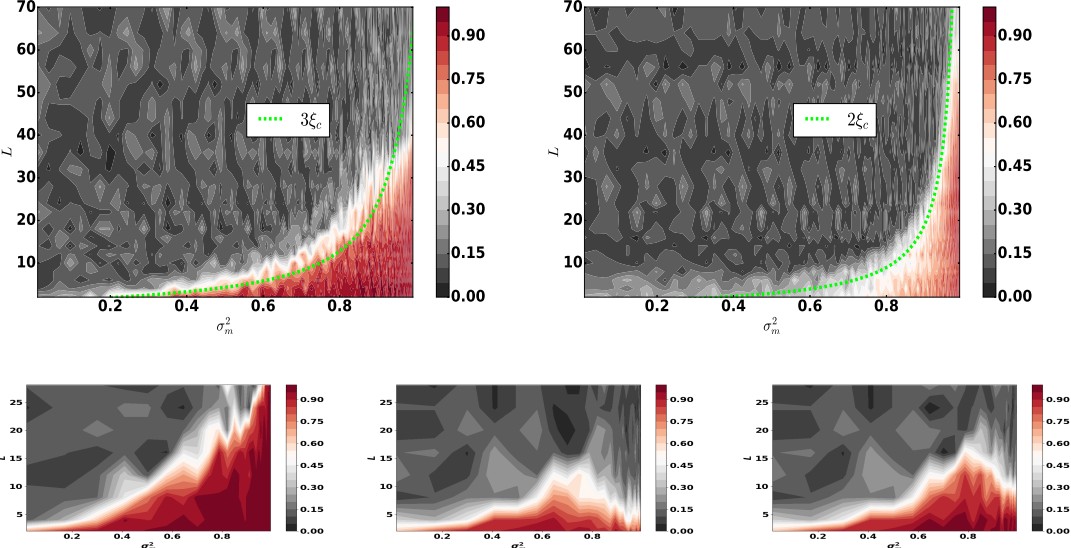

Figure 2: **Top**: Training performance of the deterministic surrogate (left) and the LRT surrogate for stochastic binary weights and continuous neurons (right). The vertical axis represents network depth against the variance of the means $\sigma_m^2$. Both surrogates were trained with $\sigma_b^2 = 0$. Thus, as $\sigma_m^2 \to 1$ we approach criticality in both cases. Overlaid are curves proportional to the correlation depth scale $\xi_c$. **Bottom**: Training performance of the deterministic surrogate and its binary counterparts after training on the MNIST dataset for 5 epochs. Left: performance of the continuous surrogate. Centre: the performance of the stochastic binary network, averaged over 5 Monte Carlo samples. Right: 100 Monte Carlo samples. The deterministic binary evaluation is similar to a single Monte Carlo sample, resembling the central figure.

## 5 DISCUSSION

This study of two classes of surrogate networks, and the derivation of their initialisation theories has yielded results of practical significance. Based on the results of Section 3, in particular Claims 1-3, we can offer the following advice. If a practitioner is interested in training networks with binary weights and neurons, one should use the deterministic surrogate, not the LRT surrogate, since the latter has no critical initialisation. If a practitioner is interested in binary weights only, the LRT in this case does have a critical initialisation (and is the only choice from amongst these two classes of surrogate). Furthermore, both networks are critically initialised when $\sigma_b^2 \to 0$ and by setting the means of the weights to $\pm 1$.

It was seen that during training, when evaluating the stochastic binary counterparts concurrently with the surrogate, the performance of binary networks was worse than the continuous model, especially as depth increases. We reported that the stochastic binary network, with more samples, outperformed the deterministic binary network, a reasonable result since the objective optimised is the expectation over an ensemble of stochastic binary networks.

A study of random deterministic binary networks, included in the Appendices, and published recently Blumenfeld et al. (2019) for a different problem, reveals unsurprisingly that binary networks are always in a chaotic phase. However a binary network which is *trained* via some algorithm will of course have different signal propagation behaviour. It makes sense that the closer one is to the early stages of the training process, the closer the signal propagation behaviour is to the randomly initialised case. We might expect that as training progresses the behaviour of the binary counterparts approaches that of the trained surrogate. Any such difference would not be observed for a heuristic surrogate as used in Courbariaux & Bengio (2016) or Rastegari et al. (2016), which has no continuous forward propagation equations.

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

# A  Derivation of deterministic surrogate networks

## A.1  Integrating over stochastic or deterministic binary neurons

The form of each neuron's probability distribution depends on the underlying noise model. We can express a stochastic binary random variable $\mathbf{S} \in \{\pm 1\}$ with $\mathbf{S} \sim p(\mathbf{S}; \theta)$ via its latent variable formulation,

$$\mathbf{S} = \text{sign}(\theta + \alpha \mathbf{L}) \tag{22}$$

In this form $\theta$ is referred to as a "natural" parameter, from the statistics literature on exponential families. The term $\mathbf{L}$ is a latent random noise, which determines the form of the probability distribution. We also introduce a scaling $\alpha$ to control the variance of the noise, so that as $\alpha \to 0$ the neuron becomes a deterministic sign function. Letting $\alpha = 1$ for simplicity, we see that the probability of the binary variable taking a positive value is

$$p(\mathbf{S} = +1) = \int_{-\infty}^{-\theta} p(\mathbf{L}) d\mathbf{L} \tag{23}$$

where $p(\mathbf{L})$ is the known probability density function for the noise $\mathbf{L}$. The two common choices of noise models are Gaussian or logistic noise. The Gaussian of course has shifted and scaled $\text{erf}(\cdot)$ function as its cumulative distribution. The logistic random variable has the classic "sigmoid" or logistic function as its CDF, $\sigma(z) = \frac{1}{1+e^{-z}}$.

Thus, the probability of a the variable being positive is a function of the CDF. In the Gaussian case, this is $\Phi(\theta)$. By symmetry, the probability of $p(\mathbf{S} = -1) = \Phi(-\theta)$. Thus, we see the probability distribution for the binary random variable in general is the CDF of the noise $\mathbf{L}$, and we write $p(\mathbf{S}) = \Phi(\mathbf{S}\theta)$. In the logistic noise case, we have $p(\mathbf{S}) = \sigma(\mathbf{S}\theta)$

For the stochastic neurons, the natural parameter is the incoming field $\mathbf{h}_i^\ell = \sum_j \mathbf{S}_{i,j}^\ell \mathbf{x}_j^{\ell-1} + b_i^\ell$. Assuming this is approximately Gaussian in the large layer width limit, we can successively marginalise over the stochastic inputs to each neuron, calculating an approximation of each neuron's probability distribution, $\hat{p}(\mathbf{x}_i^\ell)$. This approximation is then used in the central limit theorem for the next layer, and so on.

For the case of neurons with latent Gaussian noise as part of the binary random variable model, the integration over the pre-activation field (assumed to be Gaussian) is exact. Explicitly,

$$p(\mathbf{x}_i^\ell) = \sum_{\mathbf{x}^{\ell-1}} \sum_{\mathbf{S}^\ell} p(\mathbf{x}_i^\ell | \mathbf{x}^{\ell-1}, \mathbf{S}^\ell) p(\mathbf{S}^{\ell-1}) \hat{p}(\mathbf{x}^\ell)$$

$$\approx \int \Phi(\mathbf{x}_i^\ell \mathbf{h}_i^\ell) \mathcal{N}(\mathbf{h}_i^\ell | \bar{h}_i^\ell, (\Sigma_{MF}^\ell)_{ii})$$

$$= \Phi\left(\frac{\bar{h}_i^\ell}{\sqrt{1 + 2(\Sigma_{MF}^\ell)_{ii}}} \mathbf{x}_i^\ell\right) = \hat{p}(\mathbf{x}_i^\ell) \tag{24}$$

where $\Phi(\cdot)$ is the CDF of the Gaussian distribution. We have again $\Sigma_{MF}$ denoting the mean field approximation to the covariance between the stochastic binary pre-activations. The Gaussian expectation of the Gaussian CDF is a known identity, which we state in more generality in the next section, where we also consider neurons with logistic noise.

This new approximate probability distribution $\hat{p}(\mathbf{x}_i^\ell)$ can then used as part of the Gaussian CLT applied at the next layer, since it determines the means of the neurons in the next layer,

$$\mathbb{E}\mathbf{x}_i^\ell = 2\Phi\left(\frac{\bar{h}_i^\ell}{\sqrt{1 + (\Sigma_{MF}^\ell)_{ii}}}\right) - 1 \tag{25}$$

If we follow these setps from layer to layer, we see that we are actually propagating approximate means for the neurons, combined non-linearly with the means of the weights. Given the approximately analytically integrated loss function, it is possible to perform gradient descent with respect to the means and biases, $M_{ij}^\ell$ and $b_i^\ell$.

In the case of deterministic $\text{sign}()$ neurons we obtain particularly simple expressions. In this case the "probability" of a neuron taking, for instance, positive is just Heaviside step function of the incoming field. Denoting the Heaviside with $\Theta(\cdot)$, we have

$$
\begin{aligned}
p(\mathbf{x}_i^\ell) &= \sum_{\mathbf{x}^{\ell-1}} \sum_{\mathbf{S}^\ell} p(\mathbf{x}_i^\ell | \mathbf{x}^{\ell-1}, \mathbf{S}^\ell) p(\mathbf{S}^{\ell-1}) \hat{p}(\mathbf{x}^{\ell-1}) \\
&\approx \int \Theta(\mathbf{x}_i^\ell \mathbf{h}_i^\ell) \mathcal{N}(\mathbf{h}_i^\ell | \bar{h}_i^\ell, (\Sigma_{MF}^\ell)_{ii}) \\
&\approx \Phi\left( \frac{\bar{h}_i^\ell}{(\Sigma_{MF}^\ell)_{ii}^{-\frac{1}{2}}} \mathbf{x}_i^\ell \right) = \hat{p}(\mathbf{x}_i^\ell)
\end{aligned}
\tag{26}
$$

We can write out the network forward equations for the case of deterministic binary neurons, since it is a particularly elegant result. In general we have

$$
\bar{x}_i^\ell = \phi(\eta h^\ell), \quad h^\ell = \sqrt{\Sigma_{MF}} \bar{h}^\ell, \quad \bar{h}^\ell = M^\ell x^{\ell-1} + b^\ell
\tag{27}
$$

where $\phi(\cdot) = \text{erf}(\cdot)$ is the mean of the next layer of neurons, being a scaled and shifted version of the neuron's noise model CDF. The constant is $\eta = \frac{1}{\sqrt{2}}$, standard for the Gaussian CDF to error functin conversion.

## A.2 Exact and approximate Gaussian integration of sigmoidal functions

We now present the integration of stochastic neurons with logistic as well as Gaussian noise as part of their latent variable models. The logistic case is an approximation built on the Gaussian case, motivated by approximating the logistic CDF with the Gaussian CDF. The reason we may be interested in using logistic CDFs, rather than just considering latent Gaussian noise models which integrate exactly, is not justified in any rigorous or experimental way. Any such analysis would likely consider the effect of the tails of the logistic versus the Gaussian distributions, where the logistic tails are much heavier than those of the Gaussian. One historic reason for considering the logistic function, we note, is the prevalence of logistic-type functions (such as $\tanh(\cdot)$) in the neural network literature. The computational cost of evaluating either logistic or error functions is similar, so there is no motivation from the efficiency side. Instead it seems a historic preference to have logistic type functions used with neural networks.

As we saw in the previous subsection, the integration over the analytic probability distribution for each neuron gave a function which allows us to calculate the means of the neurons in the next layer. Therefore, we directly calculate the expression for the means.

The Gaussian integral of the Gaussian CDF was used in the previous section to derive the exact probability distribution for the stochastic binary neuron in the next layer. The result is well known, and can be stated in generality as follows,

$$
\int_{-\infty}^{\infty} \Phi(ay) \frac{e^{-\frac{(y-x)^2}{2\sigma^2}}}{\sqrt{2\pi\sigma^2}} dy = \Phi\left( \frac{x}{\sqrt{1 + a^2\sigma^2}} \right)
\tag{28}
$$

We can integrate a logistic noise binary neuron using this result as well. The idea is to approximate the logistic noise with a suitably scaled Gaussian noise. However, since the overall network approximation results in propagating means from layer to layer, we can equivalently need to approximate the $\tanh(\cdot)$ with the with the erf. Specifically, if we have $f(x; \alpha) = \tanh(\frac{x}{\alpha})$, an approximation is $g(x; \alpha) = \text{erf}(\frac{\sqrt{\pi}}{2\alpha} x)$, by requiring equality of derivatives at the origin. In order to establish this, consider

$$
f'(0; \alpha) = (1 - \tanh^2(0/\alpha)) \frac{1}{\alpha} = \frac{1}{\alpha}
\tag{29}
$$

and

$$
\frac{d\,\text{erf}(x; \sigma)}{dx} \Big|_{x=0} = \frac{2}{\sqrt{\pi\sigma^2}} e^{-x^2/\sigma^2} \Big|_{x=0} = \frac{2}{\sqrt{\pi\sigma^2}}
\tag{30}
$$

Equating these, gives $\sigma^2 = \frac{4\alpha^2}{\pi}$, thus $\sigma = \frac{2\alpha}{\sqrt{\pi}}$.

The approximate integral over the stochastic binary neuron mean is then

$$\int_{-\infty}^{\infty} f(y; \alpha) \frac{e^{-\frac{(y-x)^2}{2\sigma^2}}}{\sqrt{2\pi\sigma^2}} dy \approx \int_{-\infty}^{\infty} \text{erf}(\frac{\sqrt{\pi}}{2\alpha} y) \frac{e^{-\frac{(y-x)^2}{2\sigma^2}}}{\sqrt{2\pi\sigma^2}} dy \tag{31}$$

$$= \text{erf}(\frac{\sqrt{\pi}}{2\alpha\gamma} x) \tag{32}$$

$$\text{with } \gamma = \sqrt{1 + \frac{\pi\sigma^2}{2\alpha^2}} \tag{33}$$

If we so desire, we can approximate this again with a $\tanh(\cdot)$ using the $\tanh(\cdot)$ to $\text{erf}(\cdot)$ approximation in reverse. The scale parameter of this $\tanh(\cdot)$ will be $\alpha_2 = \frac{\pi}{4\alpha\gamma}$. If $\alpha = 1$ as is standard, then

$$\text{erf}(\frac{\sqrt{\pi}}{2\gamma} x) \approx \tanh(\frac{\pi x}{4\gamma}) \tag{34}$$

## B  EQUIVALENCE OF DETERMINISTIC AND STOCHASTIC NEURONS FOR DETERMINISTIC SURROGATE

Assume a stochastic neuron with some latent noise, as per the previous appendix, with mean $\bar{x}_i^\ell = \mathbb{E}_{p(x_i)} x_i^\ell = \phi(h_i^{\ell-1})$. The field is given by

$$h_i^\ell = \frac{1}{\sqrt{2}} \frac{\sum_j M_{ij}^\ell \phi(h_i^{\ell-1}) + b_i^\ell}{\sqrt{1 + 2\sum_j [1 - (M_{ij}^\ell)^2 \phi^2(h_i^{\ell-1})]}} \tag{35}$$

We see that the expression for the variance of the field simplifies as follows,

$$q_{aa}^\ell = \mathbb{E}(h_i^\ell)^2 = \frac{1}{2} \frac{\sum_j M_{ij}^\ell \phi(h_i^{\ell-1}) + b_i^\ell}{1 + 2\sum_j [1 - (M_{ij}^\ell)^2 \phi^2(h_i^{\ell-1})]} \tag{36}$$

$$= \frac{1}{2} \frac{N(\sigma_m^2 \mathbb{E}\phi^2(h_{j,a}^{l-1}) + \sigma_b^2)}{1 + 2(N - N\sigma_m^2 \mathbb{E}\phi^2(h_{j,a}^{l-1}))} \tag{37}$$

$$= \frac{1}{2} \frac{\sigma_m^2 \mathbb{E}\phi^2(h_{j,a}^{l-1}) + \sigma_b^2}{2(1 - \sigma_m^2 \mathbb{E}\phi^2(h_{j,a}^{l-1}))} \tag{38}$$

By similar steps, we find that in the deterministic binary neuron case, we would obtain the same expression, albeit with a different scaling constant. This is easily seen by inspection of the field term in the deterministic neuron case,

$$h_i^\ell = \frac{1}{\sqrt{2}} \frac{\sum_j M_{ij}^\ell \phi(h_i^{\ell-1}) + b_i^\ell}{\sqrt{\sum_j [1 - (M_{ij}^\ell)^2 \phi^2(h_i^{\ell-1})]}} \tag{39}$$

which again was derived in the previous appendix.

## C  DERIVATION OF SIGNAL PROPAGATION EQUATIONS IN DETERMINISTIC SURROGATE NETWORKS

Here we present the derivations for the signal propagation in the continuous network models studied in the paper.

## C.1 VARIANCE PROPAGATION

We first calculate the variance given a signal:

$$q_{aa}^l = \frac{1}{N_l} \sum_i \left(h_{i,a}^l\right)^2 = E\left[\left(h_{i,a}^l\right)^2\right] \tag{40}$$

Where for us:

$$h_{i,a}^l = \frac{\sum_j m_{ij}^l \phi\left(h_{j,a}^{l-1}\right) + b_i^l}{\sqrt{\sum_j \left(1 - \left(m_{ij}^l\right)^2 \phi^2\left(h_{j,a}^{l-1}\right)\right)}} \tag{41}$$

and

$$m_{ij} \sim N\left(0, \sigma_m^2\right) b_i \sim N\left(0, N_{l-1}\sigma_b^2\right) \tag{42}$$

$$
\begin{aligned}
\mathbb{E}\left[\left(h_{i,a}^l\right)^2\right] &= \mathbb{E}\left[\left(\frac{\sum_j m_{ij}^l \phi\left(h_{j,a}^{l-1}\right) + b_i^l}{\sqrt{\sum_j \left(1 - \left(m_{ij}^l\right)^2 \phi^2\left(h_{j,a}^{l-1}\right)\right)}}\right)^2\right] = \frac{\mathbb{E}\left[\left(\sum_j m_{ij}^l \phi\left(h_{j,a}^{l-1}\right) + b_i^l\right)^2\right]}{N_{l-1} - \sum_j \left(m_{ij}^l\right)^2 \phi^2\left(h_{j,a}^{l-1}\right)} \\
&= \frac{\sum_j \sigma_m^2 \mathbb{E}\phi^2\left(h_{j,a}^{l-1}\right) + N_{l-1}\sigma_b^2}{N_{l-1}\left(1 - \frac{1}{N_{l-1}}\sum_j \left(m_{ij}^l\right)^2 \phi^2\left(h_{j,a}^{l-1}\right)\right)} = \frac{N_{l-1}\sigma_m^2 \mathbb{E}\phi^2\left(h_{j,a}^{l-1}\right) + N_{l-1}\sigma_b^2}{N_{l-1}\left(1 - \sigma_m^2 \mathbb{E}\phi^2\left(h_{j,a}^{l-1}\right)\right)} \\
&= \frac{\sigma_m^2 \mathbb{E}\phi^2\left(h_{j,a}^{l-1}\right) + \sigma_b^2}{1 - \sigma_m^2 \mathbb{E}\phi^2\left(h_{j,a}^{l-1}\right)}
\end{aligned} \tag{43}
$$

Where, $\mathbb{E}\phi^2\left(h_{j,a}^{l-1}\right)$ can be written explicitly, taking into account that $h_{j,a}^{l-1} \sim N\left(0, q_{aa}\right)$:

$$
\begin{aligned}
\mathbb{E}\left[\phi^2\left(h_{j,a}^l\right)\right] &= \int \mathcal{D}h_{j,a}^l \phi^2\left(h_{j,a}^l\right) = \int dh_{j,a}^l \frac{1}{\sqrt{2\pi \mathbb{E}\left[\left(h_{j,a}^l\right)^2\right]}} \exp\left(-\frac{\left(h_{j,a}^l\right)^2}{2\mathbb{E}\left[\left(h_{j,a}^l\right)^2\right]}\right) \phi^2\left(h_{j,a}^l\right) \\
&= \int dh_{j,a}^l \frac{1}{\sqrt{2\pi q_{aa}^l}} \exp\left(-\frac{\left(h_{j,a}^l\right)^2}{2q_{aa}^l}\right) \phi^2\left(h_{j,a}^l\right)
\end{aligned} \tag{44}
$$

We can now perform the following change of variable:

$$z_{j,a}^l = \frac{h_{j,a}^l}{\sqrt{q_{aa}^l}} \tag{45}$$

Then:

$$
\begin{aligned}
\mathbb{E}\left[\phi^2\left(h_{j,a}^l\right)\right] &= \frac{1}{\sqrt{2\pi q_{aa}^l}}\sqrt{q_{aa}^l} \int dz_{j,a}^l \exp\left(-\frac{\left(z_{j,a}^l\right)^2}{2}\right) \phi^2\left(\sqrt{q_{aa}^l}z_{j,a}^l\right) \\
&= \frac{1}{\sqrt{2\pi}} \int dz \exp\left(-\frac{z^2}{2}\right) \phi^2\left(\sqrt{q_{aa}^l}z\right) \\
&= \int \mathcal{D}z \phi^2\left(\sqrt{q_{aa}^l}z\right)
\end{aligned} \tag{46}
$$

$$q_{aa}^l = \mathbb{E}\left[\left(h_{i,a}^l\right)^2\right] = \frac{\sigma_m^2 \int \mathcal{D}z \phi^2\left(\sqrt{q_{aa}^{l-1}}z\right) + \sigma_b^2}{1 - \sigma_m^2 \int \mathcal{D}z \phi^2\left(\sqrt{q_{aa}^{l-1}}z\right)} \tag{47}$$

In the first layer, input neurons are not stochastic: they are samples drawn from the Gaussian distribution $x^0 \sim N\left(0, q^0\right)$:

### C.1.1 CORRELATION PROPAGATION

To determine the correlation recursion we start from its definition:

$$c_{ab}^l = \frac{q_{a,b}^l}{\sqrt{q_{aa}^l q_{bb}^l}}, \tag{48}$$

where $q_{ab}^l$ represents the covariance of the pre-activations $h_{i,a}^l$ and $h_{i,b}^l$, related to two distinct input signals and therefore defined as:

$$q_{ab}^l = \frac{1}{N_l} \sum_i h_{i,a}^l h_{i,b}^l = \mathbb{E}\left[h_{i,a}^l h_{i,b}^l\right]. \tag{49}$$

Replacing the pre-activations with their expressions provided in eq. (41) and taking advantage of the self-averaging argument, we can then write:

$$c_{ab}^l = \frac{\sigma_m^2 \mathbb{E}\left[\phi\left(h_{j,a}^{l-1}\right)\phi\left(h_{j,b}^{l-1}\right)\right] + \sigma_b^2}{\sqrt{q_{aa}^l\left(1 - \sigma_m^2 \mathbb{E}\left[\phi^2\left(h_{j,a}^{l-1}\right)\right]\right)}\sqrt{q_{bb}^l\left(1 - \sigma_m^2 \mathbb{E}\left[\phi^2\left(h_{j,b}^{l-1}\right)\right]\right)}}. \tag{50}$$

At this point, given that $q_{aa}^l$ and $q_{bb}^l$ quite quickly approach the fixed point, we can conveniently assume $q_{aa}^l = q_{bb}^l$. Moreover, exploiting eq.(47), we can finally write the expression for the correlation recursion:

$$c_{ab}^l = \frac{1 + q_{aa}^l}{q_{aa}^l}\frac{\sigma_m^2 \mathbb{E}\left[\phi\left(h_{j,a}^{l-1}\right)\phi\left(h_{j,b}^{l-1}\right)\right] + \sigma_b^2}{1 + \sigma_b^2}. \tag{51}$$

### C.2 DERIVATION OF THE SLOPE OF THE CORRELATIONS AT THE FIXED POINT

To check the stability at the fixed point, we need to compute the slope of the correlations mapping from layer to layer at the fixed point:

$$\begin{aligned}
\chi|_{q_*} &= \frac{\partial c_{ab}^l}{\partial c_{ab}^{l-1}} \\
&= \frac{1 + q_*}{q_*}\frac{\sigma_m^2}{1 + \sigma_b^2}\frac{\partial}{\partial c_{ab}^{l-1}}\mathbb{E}\left[\phi\left(h_{j,a}^{l-1}\right)\phi\left(h_{j,b}^{l-1}\right)\right]|_{q_*}, \\
&= \frac{1 + q_*}{q_*}\frac{\sigma_m^2}{1 + \sigma_b^2}\frac{\partial}{\partial c_{ab}^{l-1}}\int \mathcal{D}z_a \mathcal{D}z_b \phi\left(u_a\right)\phi\left(u_b\right)|_{q_*}
\end{aligned} \tag{52}$$

where we get rid of $\sigma_b$ because independent from $c_{ab}^{l-1}$. Replacing the definition of $u_a$ and $u_b$ provided in the continuous model, we can explicitly compute the derivative with respect to $c_{ab}^{l-1}$:

$$\chi = \frac{1 + q_*}{q_*}\frac{\sigma_m^2}{1 + \sigma_b^2}\left(A - B\right), \tag{53}$$

where we have defined $A$ and $B$ as:

$$A = \sqrt{q_*} \int \mathcal{D}z_a \mathcal{D}z_b \phi \left( \sqrt{q_{aa}^{l-1}} z_a \right) \phi' \left( \sqrt{q_{bb}^{l-1}} \left( c_{ab}^{l-1} z_a + \sqrt{1 - \left( c_{ab}^{l-1} \right)^2} z_b \right) \right) z_a$$

$$B = \sqrt{q_*} \int \mathcal{D}z_a \mathcal{D}z_b \phi \left( \sqrt{q_{aa}^{l-1}} z_a \right) \phi' \left( \sqrt{q_{bb}^{l-1}} \left( c_{ab}^{l-1} z_a + \sqrt{1 - \left( c_{ab}^{l-1} \right)^2} z_b \right) \right) \frac{c_{ab}^{l-1}}{\sqrt{1 - \left( c_{ab}^{l-1} \right)^2}} z_b.$$

$$(54)$$

We can focus on $B$ first. Integrating by parts over $z_b$ we get:

$$B = \sqrt{q_*} \int \mathcal{D}z_a \mathcal{D}z_b \phi \left( \sqrt{q_{aa}^{l-1}} z_a \right) \frac{\partial}{\partial z_a} \phi' \left( \sqrt{q_{bb}^{l-1}} \left( c_{ab}^{l-1} z_a + \sqrt{1 - \left( c_{ab}^{l-1} \right)^2} z_b \right) \right). \qquad (55)$$

Then, integrating by parts over $z_a$, we the get:

$$B = \sqrt{q_*} \int \mathcal{D}z_a \mathcal{D}z_b \phi \left( \sqrt{q_{aa}^{l-1}} z_a \right) \phi' \left( \sqrt{q_{bb}^{l-1}} \left( c_{ab}^{l-1} z_a + \sqrt{1 - \left( c_{ab}^{l-1} \right)^2} z_b \right) \right) z_a +$$

$$- q_* \int \mathcal{D}z_a \mathcal{D}z_b \phi' \left( \sqrt{q_{aa}^{l-1}} z_a \right) \phi' \left( \sqrt{q_{bb}^{l-1}} \left( c_{ab}^{l-1} z_a + \sqrt{1 - \left( c_{ab}^{l-1} \right)^2} z_b \right) \right). \qquad (56)$$

Replacing $A$ and $B$ in eq. (53), we then obtain the closest expression for the stability at the variance fixed point, namely:

$$\chi|_{q_*} = \frac{1 + q_*}{1 + \sigma_b^2} \sigma_m^2 \int \mathcal{D}z_a \mathcal{D}z_b \phi'(u_a) \phi'(u_b) \qquad (57)$$

## C.3 VARIANCE DEPTH SCALE

As pointed out in the main text, it should hold asymptotically that:

$$|q_{aa}^{l+1} - q_*| \sim \exp\left( -\frac{l+1}{\xi_q}, \right) \qquad (58)$$

with $\xi_q$ defining the variance depth scale. To compute it we can expand over small perturbations around the fixed point, namely:

$$q_{aa}^{l+1} = q_* + \epsilon^l$$
$$= \frac{\sigma_m^2 \int \mathcal{D}z \phi^2 \left( \sqrt{q_* + \epsilon^l} z \right) + \sigma_b^2}{1 - \sigma_m^2 \int \mathcal{D}z \phi^2 \left( \sqrt{q_* + \epsilon^l} z \right)}. \qquad (59)$$

Expanding the square root for small $\epsilon^l$, we can then write:

$$q_{aa}^{l+1} \simeq \frac{\sigma_m^2 \int \mathcal{D}z \phi^2 \left( \sqrt{q_*} z + \frac{\epsilon^l}{2\sqrt{q_*}} z \right) + \sigma_b^2}{1 - \sigma_m^2 \int \mathcal{D}z \phi^2 \left( \sqrt{q_*} z + \frac{\epsilon^l}{2\sqrt{q_*}} z. \right)} \qquad (60)$$

We can now expand the activation function $\phi$ around small perturbations and then computing the square getting rid of higher order terms in $\epsilon^l$, thus finally obtaining:

$$q_{aa}^{l+1} \simeq q_* + \frac{1 + q_*}{\sqrt{q_*}} \frac{\sigma_m^2 \int \mathcal{D}z \phi \left(\sqrt{q_*}z\right) \phi' \left(\sqrt{q_*}z\right) z}{1 - \sigma_m^2 \int \mathcal{D}z \phi^2 \left(\sqrt{q_*}z\right)} \epsilon^l \tag{61}$$

Comparing this expression with the one in eq. (59), we can then write:

$$\epsilon^{l+1} \simeq \frac{1 + q_*}{\sqrt{q_*}} \frac{\sigma_m^2 \int \mathcal{D}z \phi \left(\sqrt{q_*}z\right) \phi' \left(\sqrt{q_*}z\right) z}{1 - \sigma_m^2 \int \mathcal{D}z \phi^2 \left(\sqrt{q_*}z\right)} \epsilon^l. \tag{62}$$

Integrating by parts over $z$, we then obtain:

$$\epsilon^{l+1} \simeq \left[ (1 + q_*) \frac{\sigma_m^2 \int \mathcal{D}z \phi' \left(\sqrt{q_*}z\right) \phi' \left(\sqrt{q_*}z\right) + \int \mathcal{D}z \phi'' \left(\sqrt{q_*}z\right) \phi \left(\sqrt{q_*}z\right)}{1 - \sigma_m^2 \int \mathcal{D}z \phi^2 \left(\sqrt{q_*}z\right)} \right] \epsilon^l. \tag{63}$$

Given that it holds eq. (47), and noticing that $\chi$ evaluated at the correlation fixed point $c_* = 1$ is given by:

$$\chi|_{c_*=1} = \frac{\sigma_m^2}{1 + \sigma_b^2} (1 + q_*) \int \mathcal{D}z \left[ \phi' \left(\sqrt{q_*}z\right) \right]^2, \tag{64}$$

we can finally get:

$$\epsilon^{l+1} \simeq \left[ \chi|_{c_*=1} + \frac{\sigma_m^2 (1 + q_*)}{1 + \sigma_b^2} \int \mathcal{D}z \phi'' \left(\sqrt{q_*}z\right) \phi \left(\sqrt{q_*}z\right) \right] \frac{\epsilon^l}{1 + q_*}. \tag{65}$$

Given that we expect (58) to hold asymptotically, that is:

$$\epsilon^{l+1} \sim \exp \left( -\frac{l+1}{\xi_q} \right), \tag{66}$$

we can finally obtain the variance depth scale:

$$\xi_q^{-1} = \log (1 + q_*) - \log \left( \chi|_{c_*=1} + \frac{\sigma_m^2 (1 + q_*)}{1 + \sigma_b} \int \mathcal{D}z \phi'' \left(\sqrt{q_*}z\right) \phi \left(\sqrt{q_*}z\right) \right). \tag{67}$$

## D SUPPLEMENTARY FIGURES

### D.1 CRITICAL INITIALISATION SIMULATIONS: DETERMINISTIC SURROGATE CASE

We see in Figure 3 that the set of critical initialisations exist in the plane, for but $\sigma_b^2 > 10^{-20}$ all the corresponding mean variances $\sigma + m^2 > 1$ which is not possible.

### D.2 DEPTH SCALES

We see in Figure 4 the depth scales for the deterministic surrogate. Note the divergence as one expects following the simulations in Figure 3.

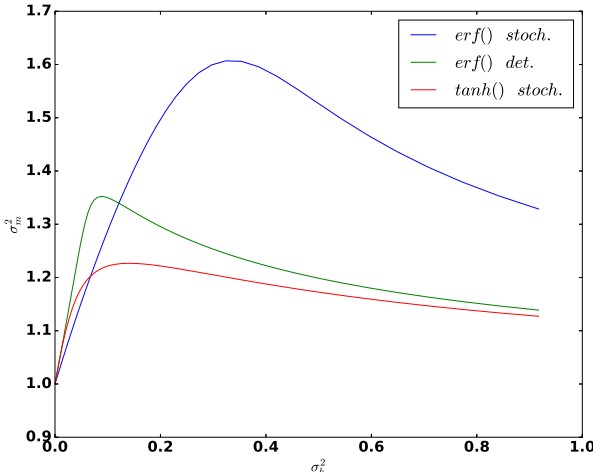

Figure 3: Plots of the valid critical initialisations for the deterministic surrogate model, for stochastic binary weights and stochastic or deterministic binary neurons. Presented are the critical initialisations in the $(\sigma_m^2, \sigma_b^2)$, for both the a) stochastic neuron case with $\phi(z) = \mathrm{erf}(\frac{1}{4} z)$, b) the deterministic sign neuron case with $\phi(z) = \mathrm{erf}(\frac{1}{2} \cdot)$, and (c) the logistic based stochastic neuron, with $\tanh()$ approximation. We see all lines are above $\sigma^2 = 1$ for all but small $\sigma_b^2 << 1$.

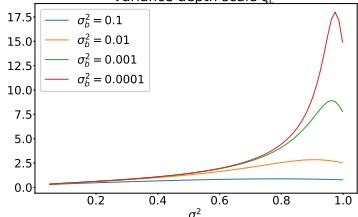
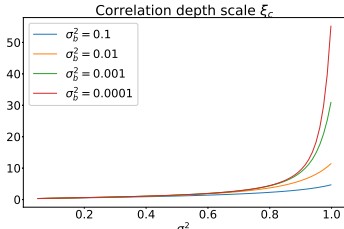

Figure 4: Depth scales as $\sigma_m^2$ is varied. (a) The depth scale controlling the variance propagation of a signal (b) The depth scale controlling correlation propagation of two signals. Notice that the correlation depth scale $\xi_c$ only diverges as $\sigma_m^2 \to 1$, whereas for standard continuous networks, there are an infinite number of such points, corresponding to various combinations of the weight and bias variances.

### D.3 Jacobian mean squared singular value and Mean Field Gradient Backpropagation

An alternative perspective on critical initialisation, to be contrasted with the forward signal propagation theory, is that we are simply attempting to control the mean squared singular value of the input-output Jacobain matrix of the entire network, which we can decompose into the product of single layer Jacobian matrices. In standard networks, the single layer Jacobian mean squared singular value is equal to the derivative of the correlation mapping $\chi$ as established in Poole et al. (2016). For the Gaussian model studied here this is not true, and corrections must be made to calculate the true mean squared singular value. This can be seen by observing the terms arising from denominator of the pre-activation field,

$$J_{ij}^{\ell} = \frac{\partial h_{i,a}^{\ell}}{\partial h_{j,a}^{\ell-1}} = \frac{\partial}{\partial h_j^{\ell}}\left(\frac{\bar{h}_{i,a}^{\ell}}{\sqrt{\Sigma_{ii}^{\ell}}}\right) = \phi'(h_{i,a}^{\ell})\Big[\frac{M_{ij}^{\ell}}{\sqrt{\Sigma_{ii}^{\ell}}} + (M_{ij}^{\ell})^2\frac{\bar{h}_{i,a}^{\ell}}{(\Sigma_{ii}^{\ell})^{3/2}}\phi(h_{i,a}^{\ell})\Big] \qquad (68)$$

Since $\Sigma_{ii}$ is a quantity that scales with the layer width $N_\ell$, it is clear that when we consider squared quantities, such as the mean squared singular value, the second term, from the derivative of the denominator, will vanish in the large layer width limit. Thus the mean squared singular value of the single layer Jacobian approaches $\chi$. We will proceed as if $\chi$ is the exact quantity we are interested in controlling. The analysis involved in determining whether the mean squared singular value is well approximated by $\chi$ essentially takes us through the mean field gradient backpropagation theory as described in Schoenholz et al. (2016). This idea provides complementary depth scales for gradient signals travelling backwards.

## E  Reparameterisation trick surrogate

### E.1  Signal propagation equations

We present, in slightly more detail, the signal propagation equations for the case of continuous neurons and stochastic binary weights yields the variance map,

$$q_{aa} = \mathbb{E}\phi^2(h_{j,a}^{l-1}) + \sigma_b^2 \qquad (69)$$

Thus, once again, the variance map does not depend on the variance of the means of the binary weights. The covariance map however does retain a dependence on $\sigma_m^2$,

$$q_{ab}^l = \sigma_m^2\mathbb{E}\phi(h_{j,a}^{l-1})\phi(h_{j,a}^{l-1}) + \sigma_b^2 \qquad (70)$$

with the same expression as before. The correlation map is given by

$$c_{ab}^l = \frac{\sigma_m^2\mathbb{E}\phi(h_{j,a}^{l-1})\phi(h_{j,a}^{l-1}) + \sigma_b^2}{\mathbb{E}\phi^2(h_{j,a}^{l-1}) + \sigma_b^2} \qquad (71)$$

and we have the derivative of the correlation map given by

$$\chi = \sigma_m^2\mathbb{E}\phi'(h_{j,a}^{l-1})\phi'(h_{j,b}^{l-1}) \qquad (72)$$

### E.2  Determining the critical initialisation conditions

We recount the argument from the paper here. Since the mean variance $\sigma_m^2$ does not appear in the variance map, we must once again consider different conditions for critical initialisation. Specifically, from the correlation map we have a fixed point $c^* = 1$ if and only if

$$\sigma_m^2 = 1 \qquad (73)$$

In turn, the condition $\chi_1 = 1$ holds if

$$\mathbb{E}[(\phi'(h_{j,a}^{l-1}))^2] = \frac{1}{\sigma_m^2} = 1 \qquad (74)$$

Thus, to find the critical initialisation, we need to find a value of $q_{aa} = \mathbb{E}\phi^2(h_{j,a}^{l-1}) + \sigma_b^2$ that satisfies this final condition. In the case that $\phi(\cdot) = \tanh(\cdot)$, then the function $(\phi'(h_{j,a}^{l-1}))^2 \leq 1$, taking the

value 1 at the origin only, this requires $q_{aa} \to 0$. Thus the critical initialisation is the singleton point $(\sigma_b^2, \sigma_m^2) = (0, 1)$. This is confirmed by experiment, as we reported in the paper.

It is of course possible to investigate this perturbed surrogate for different noise models. For example, given different noise scaling $\kappa$, as in the previous chapter, there will be a corresponding $\sigma_b^2$ that satisfy the critical initialisation condition. We leave such an investigation to future work, given the case of binary weights and continuous neurons does not appear to be of a particular interest over the binary neuron case.

# F    SIGNAL PROPAGATION OF BINARY NETWORKS

## F.1    FORWARD SIGNAL PROPAGATION

In this neural network, it should be understood that all neurons are simply $\text{sign}(\cdot)$ functions of their input, and all weights $W_{ij}^\ell \in \{\pm 1\}$ are randomly distributed according to

$$P(W_{ij}^\ell = +1) = 0.5 \tag{75}$$

$$\tag{76}$$

thus maintaining a zero mean.

The pre-activation field is given by

$$h_i^\ell = \frac{1}{\sqrt{N_{\ell-1}}} \sum_j W_{ij}^\ell \, \text{sign}(h_j^{\ell-1}) + b_i^\ell \tag{77}$$

So, the length map is:

$$q_{aa}^\ell = \int Dz (\text{sign}(\sqrt{q_{aa}^{\ell-1}} z)^2) + \sigma_b^2 \tag{78}$$

$$= 1 + \sigma_b^2 \tag{79}$$

Interestingly, this is the same value as for the perturbed Gaussian with stochastic binary weights and neurons.

The covariance evolves as

$$q_{ab}^\ell = \int Dz_1 Dz_2 \, \text{sign}(u_a) \, \text{sign}(u_b) + \sigma_b^2 \tag{80}$$

we again have a correlation map:

$$c_{ab}^\ell = \frac{\int Dz_1 Dz_2 \, \text{sign}(u_a) \, \text{sign}(u_b) + \sigma_b^2}{\sqrt{q_{aa}^{\ell-1} q_{bb}^{\ell-1}}} \tag{81}$$

where as in the paper, $u_a = \sqrt{q_{aa}^{\ell-1}} z_1$, $u_b = \sqrt{q_{bb}^{\ell-1}}(c_{ab}^{\ell-1} z_1 + \sqrt{1 - (c_{ab}^{\ell-1})^2} z_2)$.

We can find this correlation in closed form. First we rewrite our integral with $h$, for a joint density $p(h_a, h_b)$, and then rescale the $h_a$ such that the variance is 1, so that $dh_a = \sqrt{q_{aa}} dv_a$

$$\int dh_a dh_b \, \text{sign}(h_a) \, \text{sign}(h_b) p(h_a, h_b) = \int dv_a dv_b \, \text{sign}(v_a) \, \text{sign}(v_b) p(v_a, v_b) \tag{82}$$

$$= \big(2P(v_1 > 0, v_2 > 0) - 2P(v_1 > 0, v_2 < 0)\big) \tag{83}$$

where $p(v_a, v_b)$ is a joint with the same correlation $c_{ab}$ (which is now equal to its covariance), and the capital $P(v_1, v_2)$ corresponds to the (cumulative) distribution function. A standard result for standard bivariate normal distributions with correlation $\rho$,

$$P(v_1 > 0, v_2 > 0) = \frac{1}{4} + \frac{\sin^{-1}(\rho)}{2\pi}, \qquad P(v_1 > 0, v_2 < 0) = \frac{\cos^{-1}(\rho)}{2\pi} \tag{84}$$

So we then have that

$$\int dh_a dh_b \phi(h_a)\phi(h_b)p(h_a,h_b) = \sqrt{q_{aa}q_{bb}}\left(\frac{1}{2} + \frac{\sin^{-1}(c_{ab}^{\ell-1})}{\pi} - \frac{\cos^{-1}(c_{ab}^{\ell-1})}{\pi}\right) \quad (85)$$

Thus the correlation map is:

$$c_{ab}^{\ell} = \frac{\left(\frac{1}{2} + \frac{\sin^{-1}(c_{ab}^{\ell-1})}{\pi} - \frac{\cos^{-1}(c_{ab}^{\ell-1})}{\pi}\right) + \sigma_b^2}{\sqrt{q_{aa}^{\ell-1}q_{bb}^{\ell-1}}} \quad (86)$$

$$= \frac{\frac{2}{\pi}\sin^{-1}(c_{ab}^{\ell-1}) + \sigma_b^2}{\sqrt{q_{aa}^{\ell-1}q_{bb}^{\ell-1}}} \quad (87)$$

Since, from before we have $q_{aa} = 1 + \sigma_b^2$, we then obtain

$$c_{ab}^{\ell} = \frac{\frac{2}{\pi}\sin^{-1}(c_{ab}^{\ell-1}) + \sigma_b^2}{1 + \sigma_b^2} \quad (88)$$

Recall that $\sin^{-1}(1) = \frac{\pi}{2}$, so we have that $c^* = 1$ is a fixed point always.

We will now derive its slope, denoted as $\chi = \frac{\partial c_{ab}^{\ell}}{\partial c_{ab}^{\ell-1}}$, but by first integrating over the $\phi() = \text{sign}()$ non-linearities, and then taking the derivative.

Now we are in a place to take the derivative :

$$\chi = \frac{\partial c_{ab}^{\ell}}{\partial c_{ab}^{\ell-1}} = \frac{2}{\pi}\frac{1}{\sqrt{q_{aa}^{\ell-1}q_{bb}^{\ell-1}}}\frac{1}{\sqrt{1-(c_{ab}^{\ell-1})^2}} = \frac{2}{\pi}\frac{1}{(1+\sigma_b^2)}\frac{1}{\sqrt{1-(c_{ab}^{\ell-1})^2}} \quad (89)$$

We can see that the derivative $\chi$ diverges at $c_{ab}^{\ell} = 1$, meaning that there is no critical initialisation for this system. This of course means that correlations will not propagate to arbitrary depth in deterministic binary networks, as one might have expected.

## F.2 Stochastic weights and neurons

We begin again with the variance map,

$$q_{aa}^{l} = \mathbb{E}\left[(h_{i,a}^{l})^2\right] \quad (90)$$

where in this the field is given by

$$h_{i,a}^{l} = \frac{1}{\sqrt{N}}\sum_j W_{ij}^{l}x_{h_{j,a}^{l-1}} + b_i^{l} \quad (91)$$

where $x_{h_{j,a}^{l-1}}$ denotes a stochastic binary neuron whose natural parameter is the pre-activation from the previous layer.

The expectation for the length map is defined in terms of nested conditional expectations, since we wish to average over all random elements in the forward pass,

$$q_{aa}^{\ell} = \mathbb{E}_h \mathbb{E}_{x|h} x_{h_{j,a}^{l-1}} + \sigma_b^2 \quad (92)$$

$$= 1 + \sigma_b^2 \quad (93)$$

Once again, this is the same value as for the perturbed Gaussian with stochastic binary weights and neurons.

Similarly, the covariance map gives us,

$$q_{ab}^{l} = \mathbb{E}\left[h_{i,a}^{l}h_{i,b}^{l}\right] \quad (94)$$

$$= \mathbb{E}_{h_a,h_b}\mathbb{E}_{x_b|h_a}\mathbb{E}_{x_b|h_b} x_{h_{j,a}^{l-1}}x_{h_{j,b}^{l-1}} + \sigma_b^2 \qquad = \mathbb{E}\phi(h_{j,a}^{l-1})\phi(h_{j,a}^{l-1}) + \sigma_b^2 \quad (95)$$

with $phi(\cdot)$ being the mean function, or a shifted and scaled version of the cumulative distribution function for the stochastic binary neurons, just as in previous Chapters. This expression is equivalent to the perturbed surrogate for stochastic binary weights and neurons, with a mean variance of $\sigma_m^2 = 1$. Following the arguments for that surrogate, no critical initialisation exists.

### F.3 STOCHASTIC BINARY WEIGHTS AND CONTINUOUS NEURONS

In this case, as we show in the appendix, the resulting equations are

$$q_{aa}^\ell = \mathbb{E}\phi^2(h_{j,a}^{l-1}) + \sigma_b^2 \tag{96}$$

$$q_{ab}^l = \mathbb{E}\phi(h_{j,a}^{l-1})\phi(h_{j,a}^{l-1}) + \sigma_b^2 \tag{97}$$

which are, once again, the same as for the perturbed surrogate in this case, with $\sigma_m^2 = 1$. This means that this model *does* have a critical initialisation, at the point $(\sigma_m^2, \sigma_b^2) = (1, 0)$.

### F.4 CONTINUOUS WEIGHTS AND STOCHASTIC BINARY NEURONS

Similar arguments to the above show that the equations for this case are exactly equivalent to the perturbed surrogate model. This means that no critical initialisation exists in this case either.

## G MISCELLANEOUS COMMENTS

### G.1 REMARK: VALDITY OF THE CLT FOR THE FIRST LEVEL OF MEAN FIELD

A legitimate immediate concern with initialisations that send $\sigma_m^2 \to 1$ may be that the binary stochastic weights $\mathbf{S}_{ij}^\ell$ are no longer stochastic, and that the variance of the Gaussian under the central limit theorem would no longer be correct. First recall the CLT's variance is given by $\text{Var}(\mathbf{h}^\ell) = \sum_j (1 - m_j^2 x_j^2)$. If the means $m_j \to \pm 1$ then variance is equal in value to $\sum_j m_j^2(1 - x_j^2)$, which is the central limit variance in the case of only stochastic binary neurons at initialisation. Therefore, the applicability of the CLT is invariant to the stochasticity of the weights. This is not so of course if *both* neurons and weights are deterministic, for example if neurons are just $\tanh()$ functions.

