# OpenReview forum: "Critical initialisation in continuous approximations of binary neural networks"
_ICLR.cc/2020/Conference — Accept (Poster)_

### Official Review · AnonReviewer2 · 2019-10-24
**Official Blind Review #2**

**Rating:** 3

**Review:**

The paper addresses a very important and relevant topic of initialisation of weights of neural networks. It builds up on highly celebrated results of Poole, Schoenholz and others, using the language of mean field theory and an approach rooted in Dynamical Systems.
What the authors propose is an extension of the approach to other settings. The paper is very scientific and math-heavy. A good practice in such cases is to adhere to a format of a scientific mathematical paper and organise the material using Theorema, Lemmata, Propositions and Corollaries , Definitions and Proofs. Such language and framework exists for a reason - to structure the material and make the paper readable. The paper as is a stream of equations and discussion making it very unclear what the point is.
In order for this paper to be suitable for publication the reviewer would like to strongly suggest:
- Organise the material in a way that would make it clear what is claimed, what is proven etc.
- Make more specific what the added value of the paper is.
Also - for the contribution of the paper to be less incremental it would be valuable to add more formality to the original results, for example - Gaussian approximation is claimed without any sort of verification of assumptions of any version of CLT or Law of Large Numbers.

**Experience Assessment:**

I have published in this field for several years.

**Review Assessment: Checking Correctness Of Derivations And Theory:**

I assessed the sensibility of the derivations and theory.

**Review Assessment: Checking Correctness Of Experiments:**

I did not assess the experiments.

**Review Assessment: Thoroughness In Paper Reading:**

I read the paper at least twice and used my best judgement in assessing the paper.

---

> ### Author Response · Authors · 2019-11-13
> **Thank you**
>
>
> Thank you for your comments, and specific advice.
>
> We agree with the emphasis on the language of mathematical papers, and have followed your advice, including in the revision Definitions, Assumptions, Claims and Proofs.
>
> We hope this reorganisation of sections 2 and 3 under this advice have made the paper more readable.
>
> We also hope you find that 'value added' to be very clear from the revised introduction (at a big picture) and that throughout the paper the the original contributions to be more clear.
>
> As a final remark, we thank the reviewer for their concrete advice. This prompted us to reconsider several contributions that were originally downplayed in the first submission. Briefly these are:
> - novel derivations of both surrogates based on Markov chain representation
> - new reparameterisation trick surrogate for stochastic binary weights *and* neurons
> - correct backpropagation for the deterministic surrogate (not truncated as in Soudry et al.)
> - derivation of critical initialisation for continuous neuron-stochastic binary weight networks

---

### Official Review · AnonReviewer4 · 2019-11-04
**Official Blind Review #4**

**Rating:** 6

**Review:**

In this paper, the authors investigate the training dynamics of binary neural networks when using continuous surrogates for training. In particular, they study what properties the network should have at initialisation to best train. They do so via mean field approximations in the limit of very wide networks.

The authors provide concrete advice: the mean of the stochastic weights should be close to +/-1 at initialisation. Being able to give such advice to ML practitioners is of great value, but since this advice feels counter intuitive (naively, the mean of a binary +/-1 activation is typically the output of a tanh, and initialising this close to +/-1 means that gradient are going to be roughly 0, and can easily be exactly 0 in low precision computes). Right now, the justification of this initialisation is hard to find in the paper. This point would deserve a dedicated section, giving a summary of the argument, with references to more technical parts of the paper.

The presentation should also be improved as we can find the following issues:
* Missing letters, repeated words or even missing figures (Fig 5 and 7 in the appendix).
* Authors need to explain what is Edge of Chaos and give some reference (for example  C. G. Langton. Computation at the edge of chaos. Physica D, 42, 1990). This will make the paper more accessible to researchers less familiar with the theory but interested in its practical applications.
* Please, better explain figures, for example Figure 1: need to explain better what this plot is. Why are there both dotted and solid lines? What are the shaded regions: is that some confidence interval? But which one, and how many experiments were used for those plots?

**Experience Assessment:**

I do not know much about this area.

**Review Assessment: Checking Correctness Of Derivations And Theory:**

I did not assess the derivations or theory.

**Review Assessment: Checking Correctness Of Experiments:**

I assessed the sensibility of the experiments.

**Review Assessment: Thoroughness In Paper Reading:**

I read the paper at least twice and used my best judgement in assessing the paper.

---

> ### Author Response · Authors · 2019-11-13
> **Thank you**
>
> Thank you for your comments and specific issues raised.
>
> With regards to the following comments:
>
> "The authors provide concrete advice: the mean of the stochastic weights should be close to +/-1 at initialisation. Being able to give such advice to ML practitioners is of great value, but since this advice feels counter intuitive (naively, the mean of a binary +/-1 activation is typically the output of a tanh, and initialising this close to +/-1 means that gradient are going to be roughly 0, and can easily be exactly 0 in low precision computes)."
>
> - You are correct that the advice is to initialise the *means* of the stochastic *weights* close to +/-1. Your intuition on activations, such as tanh, saturating and thus producing zero gradients is also right.
>  However, we do *not* initialise activations!
> The theory developed by Poole et al. and Schoenholz et al. actually addresses the saturation of activations, in a more general framework - either looking at forward propagation (we stay (at initialisation) on the linear parts of the activation), or the equivalet of controlling the Jacobian's average (squared) singular values  - if their average =1 (= chi_1), this means there can be no saturation or zero gradient.
> Critical initialisations means precisely this setting (chi_1 =1) .
>
> "Right now, the justification of this initialisation is hard to find in the paper. This point would deserve a dedicated section, giving a summary of the argument, with references to more technical parts of the paper"
>
> -Thank you for raising this, it is an important point, and a good suggestion.
> - In the opening of section 3 we have described this idea, pointing to Claims 1 and 3, which establish the +/-1 initialisation of the weights' means.
> - We also emphasise this advice in the introduction and discussion. The paper also has an outline in the introduction which points to these sections.
> - More intuitive explanations (as discussed in our response to your question) are provided in Poole and Schoenholz, and we expect readers would follow this up. Unfortunately we find we are limited for space, otherwise we would include it. We rate these explanations quite highly, and agree with your concern.
>
>
> ***Response to specific issues:***
>
> ''Missing letters, repeated words or even missing figures (Fig 5 and 7 in the appendix)."
> - we agree, and have tidied up the paper in this regard.
>
> ''Authors need to explain what is Edge of Chaos and give some reference"
> - We have removed the phrase ''edge of chaos", since it is equivalent to the set of (sigma_m, sigma_b) which critically initialise the network. This is clear from Definition 1, and claims 1,2,3, in section 3 of the paper.
> For further information, the term 'edge of chaos' refers to the marginal stability of the c*=1 fixed point of the correlation map c(c^{l-1}) -figure 1.c). If chi=1 at c*=1, the fixed point is marginally stable. This perspective is discussed in Poole et al, Schoenholz et al. and more recently in Hayou et al. We include this discussion in the appendix.
>
>
> ''Please, better explain figures, for example Figure 1: need to explain better what this plot is. Why are there both dotted and solid lines? What are the shaded regions: is that some confidence interval? But which one, and how many experiments were used for those plots?"
>
> - The main text now describes the dotted lines (empirical means) and solid line (theory), and the shaded region corresponding to the simulations falling under one standard deviation.

---

> > ### Comment · AnonReviewer4 · 2019-11-14
> > **Thank you**
> >
> > Thank you for revisiting the paper. While it has indeed improved, I believe my original rating still applies and I've decided to retain it.

---

### Official Review · AnonReviewer5 · 2019-11-06
**Official Blind Review #5**

**Rating:** 6

**Review:**

The paper provides an in-depth exploration of stochastic binary networks, continuous surrogates, and their training dynamics with some potentially actionable insights on how to initialize weights for best performance. This topic is relevant and the paper would have more impact if its structure, presentation  and formalism could be improved. Overall it lacks clarity in the presentation of the results, the assumptions made are not always clearly stated and the split between previous work and original derivations should be improved.

In particular in section 2.1, the author should state what exactly the mean-field approximation is and at which step it is required (e.g. independence is assumed to apply the CLT but this is not clearly stated). Section 3 should also clearly state the assumptions made. That section just follows the “background” section where different works treating different cases are mentioned and it is important to restate here which cases this paper specifically considers. Aside from making assumptions clearer, it would be helpful to highlight the specific contributions of the paper so we can easily see the distinctions between straightforward adaptations of previous work and new contributions.

Specific questions:

It might be worth double checking the equation between eq. (2) and eq. (3) , the boundary case (l=0) does not make sense to me, in particular what is S^0 ?.

What does the term hat{p}(x^l) mean in the left hand side of eq.(3)?

In eq. (7) (8) why use the definition symbol := ?

At the beginning of section 3.1, please indicate what “matcal(M)” precisely refers to. Using the term P(mathcal(M) = M_ij) does not make much sense if the intent is to use a continuous distribution for the means.

Just after eq. (9), please explain what Xi_{c*} means.

Small typo: Eq. (10) is introduced as “can be read from the vector equation 31”, what is eq. (31)?

In section 5.2, why reducing the training set size to 25% of MNIST?


**Experience Assessment:**

I do not know much about this area.

**Review Assessment: Checking Correctness Of Derivations And Theory:**

I assessed the sensibility of the derivations and theory.

**Review Assessment: Checking Correctness Of Experiments:**

I assessed the sensibility of the experiments.

**Review Assessment: Thoroughness In Paper Reading:**

I read the paper at least twice and used my best judgement in assessing the paper.

---

> ### Author Response · Authors · 2019-11-13
> **Thank you**
>
> Thank you for your comments and specific questions.
>
> We have revised the paper considerably, in particular the Introduction and Sections 2 and 3. We have also included updated experiments.
>
> All assumptions are clearly stated, and justified directly. For clarity, we have also reduced the number of equations.
>
> In terms of contributions, we highlight these in the new introduction:
>
> - we have reconsidered the novelty of our Markov chain based derivation of *both* surrogates, since it is considerably simpler than previous works, and since it also allows for local-reparameteristion-trick (LRT) to be used for networks with stochastic weights *and* neurons. This is a new algorithm.
> - Also, we make clear the limitations of previous works, eg. Soudry et al. did not backpropagate correctly, ignoring all 'variance terms' (the denominators in Eq 11).
>
> ***Response to specific questions:***
> ''It might be worth double checking the equation ..."
> - thank you for picking this up, this is now Equation (4) in the updated paper. There is no S^0, the matrices S^l range from $l =1,2,3..,L$.
>
> ''In eq. (7) (8) why use the definition symbol := ?"
> -we have removed this notation.
>
> ''At the beginning of section 3.1, please indicate what “matcal(M)” precisely refers to"
> - We agree the previous notation was poor. This is now in section 3.2, and we believe we have made this much clearer.
>
> ''Just after eq. (9), please explain what Xi_{c*} means. "
> - This is now after Equation 10. We have cleared defined this notation, as well as using it in Definition 1.
>
> ''Small typo:..."
> - thank you.
>
> ''In section 5.2, why reducing the training set size to 25% of MNIST?"
> - Since we study trainability, we wish only for the neural network to fit a training set - we are unconcerned with overfitting. Our new experiments run on MNIST 50%, but our computational resources are limited.

---

### Decision · Program_Chairs · 2019-12-19

**Decision:**

Accept (Poster)

**Comment:**

The authors study neural networks with binary weights or activations, and the so-called "differentiable surrogates" used to train them.
They present an analysis that unifies previously proposed surrogates and they study critical initialization of weights to facilitate trainability.

The reviewers agree that the main topic of the paper is important (in particular initialization heuristics of neural networks), however they found the presentation of the content lacking in clarity as well as in clearly emphasizing the main contributions.
The authors imporved the readability of the manuscript in the rebuttal.

This paper seems to be at acceptance threshold and 2 of 3 reviewers indicated low confidence.
Not being familiar with this line of work, I recommend acceptance following the average review score.